# Tailoring and Long-Term Preservation of the Properties of PLA Composites with “Green” Plasticizers

**DOI:** 10.3390/polym14224836

**Published:** 2022-11-10

**Authors:** Marius Murariu, Yoann Paint, Oltea Murariu, Fouad Laoutid, Philippe Dubois

**Affiliations:** 1Laboratory of Polymeric and Composite Materials, Materia Nova Materials R&D Center & UMons Innovation Center, 3 Avenue Copernic, 7000 Mons, Belgium; 2Laboratory of Polymeric and Composite Materials, Center of Innovation and Research in Materials and Polymers (CIRMAP), University of Mons (UMons), Place du Parc 20, 7000 Mons, Belgium

**Keywords:** poly(lactic acid) PLA, composites, calcium sulfate, anhydrite II (AII), plasticizer, tributyl citrate (TBC), reactive extrusion (REX), chain extender (CE), thermal and mechanical properties, toughness, melt rheology, film aging

## Abstract

Concerning new polylactide (PLA) applications, the study investigates the toughening of PLA–CaSO_4_ β-anhydrite II (AII) composites with bio-sourced tributyl citrate (TBC). The effects of 5–20 wt.% TBC were evaluated in terms of morphology, mechanical and thermal properties, focusing on the enhancement of PLA crystallization and modification of glass transition temperature (T_g_). Due to the strong plasticizing effects of TBC (even at 10%), the plasticized composites are characterized by significant decrease of T_g_ and rigidity, increase of ductility and impact resistance. Correlated with the amounts of plasticizer, a dramatic drop in melt viscosity is also revealed. Therefore, for applications requiring increased viscosity and enhanced melt strength (extrusion, thermoforming), the reactive modification, with up to 1% epoxy functional styrene–acrylic oligomers, was explored to enhance their rheology. Moreover, larger quantities of products were obtained by reactive extrusion (REX) and characterized to evidence their lower stiffness, enhanced ductility, and toughness. In current prospects, selected samples were tested for the extrusion of tubes (straws) and films. The migration of plasticizer was not noted (at 10% TBC), whereas the mechanical and thermal characterizations of films after two years of aging evidenced a surprising preservation of properties.

## 1. Introduction

Nowadays, polylactide or polylactic acid (PLA), is among the most important biopolymers when considering further developments and the growth of global production capacities, because it is produced from renewable resources, by the fermentation of polysaccharides or sugar, such as those extracted from corn or sugar beet, and corresponding wastes [1,2,3,4]. Furthermore, the most relevant end-life scenario for PLA is related to its biodegradability under controlled industrial composting conditions, under home-composting conditions in the presence of adapted enzymes or in the natural environment (e.g., at a lower rate in soil) [5,6]. The life cycle of PLA demonstrates that this biopolymer presents a performant and sustainable alternative to petrochemical polymers, with less greenhouse gas emissions, while at the end of its service life PLA can be degraded to CO_2_ and biomass, facilitating the reduction of landfill volumes [7,8].

It is worth mentioning that the dominant end-use sector for PLA is packaging [9,10,11], followed by textile fibers [12], whereas, more recently, adequately modified PLA was considered for use in durable/engineering applications [13,14,15,16,17,18]. It finds higher added value and remains of great interest for biomedical applications, due to its biocompatibility and proprieties of biodegradation/bioresorption [19]. Moreover, PLA is currently the market leader in the segment of biobased and biodegradable plastics, and, at the same time, it is considered to be the polymer that comes closest to conventional plastics in terms of performance and production costs [20].

PLA is considered to be a key bioplastic, with the largest market significance, because it is characterized by many interesting properties (high tensile strength and modulus of elasticity, good flexural strength, optical transparency, biodegradability, among others) [2]. On the other hand, unfortunately, PLA has some shortcomings which limit its larger application: poor toughness/high brittleness, lack of elongation and flexibility, relatively low melt strength, slow crystallization rate, low heat distortion temperature (HDT), etc. [13]. Therefore, new PLA-based products (composites, nanocomposites, alloys, etc.) with improved characteristics and specific end–use properties are needed to fulfill the requirements of different sectors. The properties of PLA are enhanced by combining the polyester matrix with micro- and nano-fillers, reinforcing fibers, impact modifiers, plasticizers, other polymers, and various types of additives [1,2,3,13].

The blending of PLA with mineral fillers, where the dispersed phase has dimensions from nanometers to several microns or more, can be an interesting solution to reduce its cost and to improve some specific properties, such as the rigidity, heat deflection temperature, processability, barrier performances, and so on. Following different objectives, PLA was melt-mixed with clays [13,21], CaCO_3_ [22], talc [23,24], kaolin [25], and other mineral fillers, typically used in the industry of polymer composites. On the other hand, it is important to mention that not all fillers and additives have a beneficial effect on all PLA properties. Unfortunately, some parameters are improved, whereas others are not, and, in some cases, advanced degradation of PLA, leading to a sharp reduction of thermal and mechanical performances, is reported. Therefore, to maximize the benefits and versatility of PLA products, it is necessary to understand, and combine, the relationship between the properties of the polyester matrix and characteristics of dispersed phases (micro- and nano-fillers), additives, etc., their compatibility and interactions, stabilizing or degradation effects, the influence of the manufacturing process on the final product characteristics, and so forth [2,13].

In response to the demand to extend the range of PLA applications, while reducing its production cost, we recently demonstrated that commercial PLAs can be effectively melt-blended with CaSO_4_ β-anhydrite II form (AII), a filler produced by the calcination of natural gypsum as primary raw material [26]. Obviously, these two products (i.e., PLA and AII) can lead to PLA composites characterized by remarkable thermal and mechanical properties, even in the absence of any filler surface treatments. Certainly, so-called insoluble anhydrite (AII), produced at temperatures higher than 500 °C using synthetic or natural gypsum, is less sensitive to moisture and water absorption, and, therefore, this filler was preferred for melt–mixing with polymers, such as PLA, characterized by high sensitivity to degradation by hydrolysis during processing at high temperature.

Unfortunately, the PLA–AII composites were characterized by high stiffness and low ductility, while the impact resistance was dramatically diminished by high filling, e.g., at 40 wt.% AII [2,26]. Therefore, for specific applications requiring increased flexibility, tensile/impact toughness and/or ductility (e.g., injection molded (IM) items, tubes (straws) and films/sheets produced by extrusion), the PLA–AII composites do not have the properties required for good processability and/or to permit their suitable use (e.g., to avoid breaking and cracking during fabrication, storage, and transport). Furthermore, the packaging sector requires materials which allow plastic deformation at high impact rates, together with advanced elongation [27,28].

Hence, the addition of a third component into PLA–AII composites, i.e., a plasticizer [29], an impact modifier [30], a flexible biodegradable (co)polyester [31], such as poly(butylene adipate-co-terephthalate) (PBAT), or other (co)polymers [32], etc., can represent alternatives of choice to obtain PLA composites with improved toughness [29,32,33,34].

Nevertheless, it is largely recognized that the mechanical properties of PLA and of PLA composites (i.e., their flexibility/stiffness, ductility, and impact resistance), can be tailored up with different categories of plasticizers [35,36,37,38,39,40,41]. The ideal plasticizer for PLA must be a biodegradable product, sufficiently non-volatile, leading to a substantial reduction of glass transition temperature (T_g_) and adequate mechanical properties, i.e., a significant decrease of Young’s modulus, an increase of ductility and impact resistance, with an important concern being long-term preservation of all these properties [42,43,44]. In other words, this requires the absence of plasticizer exudation/migration during prolonged aging. On the other hand, it is worth noting that the plasticizers can reduce the melt viscosity of PLA, a feature of interest for IM applications, although this is less desired for techniques of processing requiring higher melt strength and viscosity (extrusion, thermoforming).

Regarding the different plasticizers confirmed for PLA, first, it is considered that, although the monomer lactide itself is of high effectiveness, the main drawback relates to its fast migration to the material surface [45]. Several studies were conducted using different commercial or synthesized plasticizers, such as esterified oligomers of L-lactic acid [46], epoxidized fatty acid esters [47], glycerol esters [42], citrate esters [42,48], poly(ethylene glycol) [35,45,49], citrate oligoesters [36], poly(propylene glycol) [50], polymeric adipates [29,38], cardanol-derived plasticizers [49], and so on. Generally, it is agreed that the small molecules are more efficient plasticizers, and that the miscibility of PLA with a plasticizer from the same chemical family decreases with increase in the molecular weights of the plasticizer. For instance, it is deemed that the “citrates”, such as tributyl citrate (TBC), acetyl tributyl citrate (ATBC), triethyl citrate (TEC), acetyl triethyl citrate (ATEC), etc., belong among the most efficient “green” monomeric plasticizers for PLA. They are biodegradable, nontoxic, and approved for food packaging, medical applications, and the realization of toys. Their good compatibility with PLA is mostly attributed to the polar interactions between the ester groups of PLA and those of citrates [44]. TBC is one of the most effective plasticizers for PLA, produced using entirely renewable components (i.e., citric acid and n-butanol, as raw materials). Former studies showed that between 10 and 20 wt.% of plasticizer in PLA, results in a higher elongation at break and lower T_g_ compared to the unplasticized PLA [40]. TBC was tested, with promising results, for its ability to modify PLA properties (ductility and stiffness, but also impact resistance, the kinetics of PLA crystallization, etc.) [51,52,53,54,55,56]. Ljungberg et al. transesterified TBC with diethylene glycol to obtain oligomeric plasticizers of higher molecular weights. The study of the effects of these oligomers on the thermo-mechanical and aging properties of PLA showed that the oligomers did not lower the T_g_ as greatly as the monomeric TBC [36]. Hassouma et al. [57], investigated the chemical grafting of TBC on PLA by reactive extrusion (REX). The grafting reaction of TBC with maleic anhydride grafted PLA (MAG-PLA) shifted the T_g_ toward higher temperatures. Interestingly, in the case of PLA/TBC and PLA/MAG-PLA/TBC blends, no major leaching phenomenon was noticed during aging for six months. To promote some more tortuosity for plasticizers (ATBC), and to reduce/control its migration, PLA was melt-mixed with calcium carbonate and natural nanofibers (chitin nanofibrils) [58].

To the best of our knowledge, TBC or other citrates, have not yet been tested to toughen/plasticize PLA–AII composites. However, the choice of TBC in this study as key plasticizer from the category of citrates, was based on the huge amount of work reported in the literature, the previous experience and results obtained by authors for the plasticization of PLA [29,42,53], and by considering other purposes (e.g., the lower volatility, favorable chemistry, solubility parameters, etc.). Furthermore, there is an industrial interest for the larger use of these biocomposites as sustainable alternatives for specific applications i.e., the extrusion of drinking straws, packaging, mulch films, surface coverings, and so on. Accordingly, this study represents an answer to current requests regarding the production of low-cost mineral filled PLA–AII composites designed with tailored properties (i.e., improved toughness and ductility), to allow their larger utilization (e.g., to substitute petroleum-based PP (polypropylene) in the production of straws). In this contribution, the “green” plasticizing of composites was carried out, together with reactive modification, to enhance their rheology. It shows the main experimental pathways followed from laboratory scale (internal mixers) to the production by reactive extrusion (REX) of most representative composites.

First, using the internal mixers for melt compounding, PLA was blended with AII (made from natural gypsum) and TBC as a “green” plasticizer. The percentage of AII was kept at 30 wt.%, because its incorporation at higher amounts (i.e., 40% in PLA) was reported to induce dramatic decrease of impact resistance [26]. The properties of composites obtained following the addition of 5–20% TBC and 30% filler into a selected/semicrystalline PLA matrix, were deeply characterized. Here, the key objective was to benefit from plasticization by improving material ductility/toughness, while preserving a valuable rigidity/stiffness by filling with low-cost AII. It was expected that reinforcing with fillers and control of PLA crystallization could overcome some undesired effects, due to the low T_g_ of plasticized composites, regarding the dimensional stability and preservation of shape of the final products.

Furthermore, aiming at overcoming the dramatic decrease of melt viscosity by plasticization, the study proposed, as its main novelty, the addition of a reactive multifunctional chain extender (CE) into plasticized PLA-AII compositions to improve their rheology and to control the melt flow rate (MFR). From the category of additives that are of interest for PLA, epoxy functional styrene–acrylic oligomers (Joncryl^®^), are claimed to be efficient reactive compatibilizers, melt strengtheners and CEs for PLA blends [59,60,61,62,63,64], therefore they were tested to improve the rheology of plasticized compositions. Incontestably, from the torque rheometric evaluations and the analysis of MFR of plasticized composites, the possibility to increase and tailor the melt viscosity of plasticized composites by adapting the level of CE to the requirements of processing (e.g., medium to high viscosity) was experimentally proved. Finally, in the frame of current prospects, experimental trials to produce larger quantities of materials by REX and for their extrusion as tubes (straws) or films, were realized. Interestingly, the migration of plasticizer by exudation was not noted (on films containing 10 wt.% TBC), whereas the mechanical and thermal characterizations after two years of aging evidenced surprising preservation of properties.

By considering the overall performances of plasticized PLA–AII composites (ductility, tailored stiffness and rheology, impact resistance, etc.), these novel composites are proposed for further developments and production at larger scale. Nevertheless, by the careful choice of PLA matrix (semicrystalline or amorphous grade), and by tailoring the amounts of TBC and CE, they can be designed for processing by IM, extrusion, thermoforming, and 3D printing.

## 2. Materials and Methods

### 2.1. Materials

PLA 4032D (supplier NatureWorks LLC, Plymouth, MN, USA) is a PLA of high molecular weight and melt viscosity, designed for the extrusion of films and realization of blends. According to the information provided by the supplier, it is characterized by low D-isomer content (1.4%), a relative viscosity of 3.94, residual monomer of 0.14% and an MFR of 7 g/10 min (at 210 °C, 2.16 kg). The peak of melting temperature (DSC method) is at about 170 °C.

CaSO_4_ β-anhydrite II (AII), delivered as “ToroWhite” filler, was kindly supplied by Toro Gips S.L. (Zaragoza Spain). According to the information provided by the supplier, AII is obtained from selected food and pharmaceutical grades of high purity natural gypsum. It is characterized by high whiteness/lightness (L*), AII being an alternative of choice as a white pigment (TiO_2_) extender (available also as TOROWHITE Ti-ExR04). Color measurements performed in the CIELab mode (illuminate D65, 10°) with a SpectroDens Premium (TECHKON GmbH, Königstein, Germany) proved the high lightness of the AII sample, i.e., L* of about 96.0. Figure 1a,b shows selected SEM pictures to illustrate the morphology of the AII used in this study. The granulometry of AII was characterized by Dynamic Light Scattering, using a Mastersizer 3000 laser particle size analyzer (Malvern Panalytical Ltd., Malvern, UK), the microparticles having a Dv50 of 5.5 µm and a Dv90 of 15 µm.

Tributyl citrate (TBC) was supplied by Sigma-Aldrich with the following characteristics: molecular weight = 360.4 g/mol, density at 20 °C = 1.04 g/mL, purity ≥ 99%.

Joncryl^®^ ADR 4468 (supplier BASF) is a multifunctional reactive epoxy styrene–acrylate oligomeric CE, a product approved for food contact applications. It is a polymeric chain extender (CE), with low epoxy equivalent weight (high number of epoxy groups per chain), that reacts with the end groups of polyesters and increases their molecular weights, melt viscosity, and melt strength. According to the technical sheet of this product, it has a specific gravity (at 25 °C) of 1.08, a molecular weight (M_w_) = 7.250, epoxy equivalent by weight = 310 g/mol and a glass transition temperature (T_g_) of about 59 °C. It is referred to hereinafter as “Joncryl” and abbreviated to “J”. The general structures of Joncryl and TBC are shown in the Figure 1a,b.

### 2.2. Production of Plasticized Composites

#### 2.2.1. Melt Compounding with Torque Measuring Mixers

PLA and AII were dried overnight, at 70 °C and 100 °C, respectively, to limit PLA hydrolytic and thermal degradation during melt processing due to the presence of moisture. TBC was used as received. Starting from dry-mixed PLA/AII blends and 5–20% TBC, plasticized PLA composites were obtained by melt compounding at 190 °C, using a Brabender bench scale internal measuring mixer, having the ability to work as a torque rheometer (W50EHT, Plastograph EC, Brabender GmbH &. Co. KG, Duisburg, Germany) equipped with “came” blades. Conditions of processing: feeding at 30 rpm for 3 min, followed by melt mixing for 7 min at 100 rpm. PLA–30% AII composites were produced under similar conditions and used as reference. For reasons of clarity, the codes, and compositions of plasticized composites (PLA–AII–TBC), discussed in the Section 3.1, are shown in Table 1. The so obtained blends were processed by compression molding (CM) at 190 °C, using an Agila PE20 hydraulic press to obtain plates (~3.1 mm thickness). More specifically, the material was first maintained at low pressure for 180 s (3 degassing cycles), followed by a high-pressure cycle at 150 bars for 120 s. Then, the cooling was realized under pressure (50 bars) for 300 s using tap water (temperature slightly > 10 °C). The plates produced by CM were used to obtain specimens for mechanical characterizations. Throughout this contribution, all percentages are given as weight percent (wt.%).

#### 2.2.2. Reactive Melt Blending with Torque Measuring Mixers

To increase the melt viscosity of plasticized composites, and to obtain information about the effects of CE (Joncryl), PLA compositions containing up to 1% J were produced by reactive melt blending, using as equipment the internal mixer and its function of torque rheometer. The codes of these samples are abbreviated hereafter as “PLA/xJ–yTBC” and “PLA/xJ–AII–yTBC”, where x and y, are, respectively, the amounts (wt.%) of Joncryl and of plasticizer in the blends, whereas the percentage of filler in composites was kept at 30% AII. Noteworthy is the fact that the evolution of torque during reactive blending, and the values of MFR of the final blends, were mainly concerned for certifying the changes of melt viscosity (results discussed in Section 3.2).

#### 2.2.3. Reactive Extrusion Using Twin-Screw Extruders (TSE)

In the subsequent experimental step, larger quantities of selected composites were produced by REX, using a Leistritz twin-screw extruder (TSE) as equipment (ZSE 18 HP-40, L (length)/D = 40, diameter (D) of screws = 18 mm). Before melt compounding, PLA and AII filler were dried overnight at 70 °C and 100 °C, respectively, using drying ovens with recirculating hot air. The experimental setup used to produce plasticized composites in bigger quantities is shown in Figure 2. Two separate gravimetric feeders were used for the dosing of PLA/J blends (previously dry-mixed) and of filler (AII). The plasticizer (TBC as received) was introduced into TSE via a feeder pump through a special capillary system placed on zone 3 of the extruder. The parameters of REX/melt compounding were as follows: (a) temperatures on the heating zones of TSE: Z1 = 150 °C, Z2 = 180 °C, Z3 = 185 °C, Z4–Z6 = 180 °C, Z7 = 175 °C; (b) die of extrusion = 175 °C; (c) speed of the screws = 170 rpm; (d) throughput = 3 kg/h.

To characterize the samples produced by REX, after the previous drying of granules overnight, at the temperature of 70 °C, specimens for tensile, flexural and impact tests, were produced with a Babyplast 6/10 P injection molding (IM) machine, using adapted processing temperatures (e.g., Z1 = 175 °C; Z2 = 180 °C; Z3 (die) = 165 °C; temperature of the mold = 20–30 °C). Samples from a selected composition (PLA/0.3J–AII–10TBC) were evaluated in the frame of current prospects to produce tubes of small diameter (straws) and films (details hereinafter), whereas bigger quantities of granules were tested by external users specialized in the production of drinking straws.

#### 2.2.4. Extrusion Laboratory Tests to Produce Tubes (Straws) and Films

The drying of granules (at 70 °C, overnight) was followed by the extrusion of tubes and films (Figure 3a,b) using a Brabender laboratory single screw extruder (D = 19 mm, L/D = 25) equipped with special die heads (i.e., tubing die head with internal/external diameter of 2 mm/4 mm and a ribbon die of 100 mm wide with a gap of 0.5 mm). As downstream equipment for the drawing and cooling, a silicone-coated conveyor belt was used to take the tubes. Temperatures on the heating zones of the extruder for the realization of tubes were as follows: Z1 = 165 °C, Z2 = 175 °C, Z3 = 170 °C, Z4 = 165 °C; temperature of the die head = 160 °C. On the other hand, for the extrusion of films of different thicknesses (0.1–0.5 mm), a Brabender Univex downstream device for draw-off, cooling and winding of flat films was used. The following temperatures of extrusion were utilized: Z1 = 165–170 °C, Z2 = 185 °C, Z3 = 185 °C, Z4 = 185 °C, extrusion head = 180 °C.

### 2.3. Methods of Characterization

(a) Thermogravimetric analyses (TGAs) were performed using a TGA Q50 (TA Instruments, New Castle, DE, USA) by heating the samples under air from room temperature (RT) up to a maximum of 800 °C (platinum pans, heating ramp of 20 °C/min, 60 cm^3^/min air flow).

(b) Differential Scanning Calorimetry (DSC) measurements were accomplished by using a DSC Q200 from TA Instruments (New Castle, DE, USA) under nitrogen flow. The traditional DSC procedure was as follows: first, heating scan at 10 °C/min from 0 °C up to 200 °C, isotherm at this temperature for 2 min, and then, cooling by 10 °C/min to −20 °C, and, finally, a second heating scan from −20 to 200 °C at 10 °C/min. The first scan was used to erase the prior thermal history of the polymer samples. The events of interest linked to the crystallization of PLA during DSC cooling scans, i.e., the crystallization temperature (T_c_) and the enthalpies of crystallization (ΔH_c_), were quantified using TA Instruments Universal Analysis 2000 software (Version 3.9A (TA Instruments—Waters LLC, New Castle, DE, USA)). Noteworthy is the fact that all data were normalized to the amounts of PLA from the samples. The thermal parameters were also evaluated in the second DSC heating scan and abbreviated as follows: glass transition temperature (T_g_), cold crystallization temperature (T_cc_), enthalpy of cold crystallization (ΔH_cc_), melting peak temperature (T_m_), melting enthalpy (ΔH_m_), and final DC (χ). The DC (degree of crystallinity) was determined using the following general equation:(1)$$\chi \;=\frac{\left(\Delta H_{m}\;-\;\Delta H_{\mathrm{cc}}\right)\;}{\Delta H_{m}^{0}\times \;W_{\mathrm{PLA}}}\times 100\;\left(\%\right)$$
where ΔH_m_ and ΔH_cc_ are the enthalpies of melting and of cold crystallization, respectively, W is the weight fraction of PLA in composites, and $\Delta H_{m}^{0}$ is the melting enthalpy of 100% crystalline PLA considered 93 J/g [26]. Notably, the DC was calculated by subtracting the enthalpy of cold crystallization (∆H_cc_) and of pre-melt crystallization (if it was evidenced on DSC curves), from the melting enthalpy (ΔH_m_). Still, to characterize the changes of thermal properties of films during aging of up to 2 years, only the results obtained in the first DSC scan were considered. 

(c) Mechanical testing: Tensile tests were performed with a Lloyd LR 10K bench machine (Lloyd Instruments Ltd., Bognor Regis, West Sussex, UK), according to ASTM D638-02a norms on specimens-type V, at a typically speed of 10 mm/min (specimens of 3.1–3.2 mm thickness). The flexural properties were determined on selected samples using a three-point bending test and NEXYGEN program (Lloyd Instruments Ltd.). The measurements were performed on a minimum of five rectangular specimens (63 × 12 × 3.2 mm^3^) by using a Lloyd LR 10K tensile bench adapted with bending grips (span = 50 mm), in accordance with ISO 178, at a testing speed of 2 mm/min. For the characterization of Izod impact resistance, a Ray-Ran 2500 pendulum impact tester, and a Ray-Ran 1900 notching apparatus (Ray-Ran Test Equipment Ltd., Warwickshire, UK) were used, according to ASTM D256 norm (method A, 3.46 m/s impact speed, 0.668 kg hammer). All mechanical tests were carried out on specimens previously conditioned for at least 48 h at 20 ± 2 °C under a relative humidity of 50 ± 3%, and the values were averaged over at least five measurements.

(d) Scanning Electron Microscopy (SEM) analyses on previously cryofractured samples at a liquid nitrogen temperature were performed using a FE-SEM SU-8020 Hitachi instrument (Hitachi, Tokyo, Japan), at various accelerated voltages and magnitudes. For better insight and easy interpretation, the SEM was equipped with detectors for both secondary (SE) and back-scattered electron (BSE) imaging. Reported microphotographs represented typical morphologies as observed at, at least, three distinct locations. SEM analyses (SE mode) were also performed on the surfaces of selected specimens fractured by tensile or impact testing to have more information about their behavior under the different conditions of mechanical solicitation.

(e) Rheological Measurements: The evolution of mechanical torque during melt mixing was followed using the function of torque rheometer of the internal mixer, and considered as primary rheological information in relation to the evolution of melt viscosity. The melt flow rate (MFR) was determined following the procedure described in ASTM D1238, using a Davenport 10 Melt Flow Indexer (AMETEK Lloyd Instruments Ltd., West Sussex, UK) at a temperature of 190 °C, with a 2.16 kg load.

(f) Characterization of aged films: The modification of properties after 2 years of aging was conducted by comparing the initial mechanical and thermal characteristics of extruded films to that of aged samples (storage under normal room conditions). The tensile tests were performed at a testing speed of 50 mm/min, on specimens (ASTM D638-02a, type V) obtained by cutting from films.

## 3. Results and Discussion

### 3.1. Effects of TBC Addition on the Properties of Composites

In the first step of the study, a composition reference (i.e., PLA with 30% AII) was modified by melt blending with up to 20% TBC, using the internal measuring mixer, and, subsequently, the plasticized composites were characterized to evidence their properties.

#### 3.1.1. Rheology: Evolution of Torque during Melt Mixing

Due to their “lubricating” role, plasticizers usually decrease the melt viscosity [24,66] and melt strength of PLA, and this may adversely affect the processing by extrusion (e.g., as tubes, sheets, films, etc.). Undoubtedly, by considering the evolution of mechanical torque during the melt mixing process as primary rheological information (Figure 4), it was found that the rise of TBC amounts led to the important (at 5–10% TBC) or dramatic decrease of melt viscosity of composites (at 15–20% TBC). However, the characterization of MFR of plasticized composites (Table 2) demonstrated that, by plasticizing, their melt fluidity severely increased (e.g., ~53 g/10 min was reached adding 15% TBC). This could be an important drawback for the processing of plasticized composites by extrusion, an aspect avoided or less considered in previous studies.

#### 3.1.2. Morphology of Composites

To obtain information about the extent of filler distribution and dispersion within the PLA matrix (without/with TBC), SEM images recorded over cryofractured surfaces were obtained on selected samples using back-scattered electrons (BSE) to get a higher phase contrast for AII microparticles (Figure 5a–f). The BSE technique shows high sensitivity to the differences in atomic number, giving information about composition/AII distribution (i.e., presence of Ca atoms, evidenced by the brighter zones). As was mentioned in the experimental part, the filler used for these tests was characterized by a volume median diameter of ~5 µm (analysis of granulometry by DLS). Regarding the morphology of composites, it appeared that the big aggregates are missing, whereas well distributed AII microparticles with various geometries were evidenced in the superficial zones of the cryofractured samples. Accordingly, from the SEM pictures (low magnification) it was observed that the dispersion state of AII is correct, while the differences between the “brittle” and “ductile” behaviors, ascribed respectively to the composites without/with plasticizer (Figure 5a,b vs. Figure 5c–f), were less evident, due to the preparation of samples for SEM by cryofracturing, at liquid nitrogen temperature.

Still, from the SEM pictures at higher magnification, it was evident that the great majority of AII particles featured a low aspect ratio and irregular shape of micrometric size. It is also worth noting that the plasticizer in polymeric composites can have different functions, such as for the modification of melt viscosity and lubrication of the compound, or can even contribute to better filler dispersion. Still, the use of fillers can reduce the migration of plasticizer thanks to the creation of tortuosity that forces the plasticizer molecules to follow a longer path to leave the polymeric structure. Secondly, an adsorption mechanism of the plasticizer on the surface of fillers can also be discussed [58]. Here, based on the SEM pictures (Figure 5a,b compared to Figure 5c–f) and considering that the blends were produced with internal mixer, it was difficult to claim significant improvements in the morphology of composites following the addition of plasticizer. However, the assumption in relation to the better quality of the dispersion of AII microparticles through the plasticized PLA matrix was partially supported by the additional SEM analyses, e.g., on the fractured surfaces obtained during tensile and impact testing (details elsewhere, and in the Appendix A).

#### 3.1.3. Thermal Properties: DSC and TGA

DSC is one of the most preferred methods of analysis used to highlight the effects of plasticizers. The addition of a plasticizer to a polymer generally causes an increase in free volume and lowers the glass transition temperature (T_g_). Since the diminishing of T_g_ is an excellent indicator of polymer structure and chain mobility, the plasticizing efficiency is evaluated following the decrease of T_g_ as a function of plasticizer amount.

Figure 6a,b shows the comparative DSC curves of composites with/without TBC as they were obtained, respectively, following the cooling and second heating scan at 10 °C/min (the DSC data are summarized in the Table 3). First, as is observed from the direct comparison of cooling scans, the composites containing 10–20% TBC showed advanced enthalpies of crystallization (seen at T_c_ in the ranges 75–82 °C), while in the case of unplasticized composite (i.e., PLA–AII) this thermal event was less evident (T_c_ at 95 °C). Still, the enthalpies of crystallization were clearly correlated with the plasticizer content, whereas some synergies with the filler (AII), with effects in boosting the crystallization kinetics of PLA, are to be considered. However, it was reported elsewhere that the filler (AII) can have beneficial effects on the crystallization of PLA of high L–enantiomer purity [26]. On the other hand, many papers suggest that the degree of crystallinity (DC) of PLA is markedly enhanced by the incorporation of plasticizers [67], with a role in promoting chains mobility, or following their synergies with fillers, e.g., talc [24].

The high DC achieved during the cooling process following PLA crystallization from the molten state was confirmed in the succeeding DSC heating scans (Figure 6b**,** data shown in Table 3). In fact, the samples plasticized with 10–20% TBC did not show any evidence of so-called cold crystallization, whereas they were typically characterized by higher melting enthalpies. Indeed, the unplasticized PLA–AII composites showed a moderate DC (~14%), whereas the addition of 10–20% TBC induced a dramatical DC rise to 43–47%. Linked to the effects of TBC content, the progressive increasing of plasticizer led to some reduction of T_m_, from 170 °C (composite without plasticizer) to 158 °C (PLA–AII–20TBC). It was also noted that only the compositions without, or with a low amount of plasticizer (i.e., 5%), showed the phenomenon of cold crystallization, that was proved for the lower DC of PLA–AII and PLA–AII–5TBC samples.

Regarding the evolution of T_g_, the addition of only 5% TBC led to its decrease from about 60 °C (unplasticized PLA–AII) to about 44 °C. Unfortunately, at a rate of cooling/heating of 10 °C/min, due to the high crystallinity of PLA (NB: the case of samples with 10–20% TBC), it was more difficult to determine, with accuracy, the T_g_ event on DSC curves. Consequently, additional DSC characterizations were performed on composites cooled at a rate of 40 °C/min. As far as the T_g_ was concerned (Table 4), the addition of 5 to 20% TBC led to a progressively diminishing T_g_ of about 30 °C, adding 10% TBC, or a remarkable reduction of T_g_ (e.g., at 15% and 20% TBC, to 17 °C and to about 3 °C, respectively). Generally, the higher the plasticizer content, the greater the chain mobility. However, at increased TBC loadings (e.g., 15–20%) saturation with plasticizer should be discussed, and, therefore, it would be difficult to limit its migration during the storage/aging of final products. On the other hand, interestingly, even though a high DSC cooling rate was used (i.e., 40 °C/min), it is important to point out that the DC of composites with 10–20% TBC remained impressive, i.e., 20–37% (Table 4).

Overall, the DSC data suggested that TBC has a key role in enhancing the mobility of PLA chains, being of high effectiveness in decreasing the T_g_ values. The co-addition of AII and TBC into a PLA matrix characterized by increased optical purity (in L–enantiomer) led to advanced properties/kinetics of crystallization. On the other hand, it was assumed that, following the crystallization process, the plasticizer would be primarily found in the amorphous part of the polyester matrix [68]. TBC would mostly accumulate in the amorphous phase because it was excluded from the growing of PLA crystals, a process also aided by the low temperature of crystallization [69]. Therefore, especially at high plasticizer amounts, it would be difficult to counteract the phase separation and/or migration of TBC.

In relation to the thermal properties as determined by TGA, it was generally expected to evidence a reduction of thermal stability of plasticized compounds, due to the volatility, at high temperature, of the monomeric plasticizers. Figure 7a,b shows the TG and DTG traces of plasticized compositions compared to the reference without plasticizer. It was found that the thermal stability of plasticized composites was greatly determined by the amounts of TBC, and this was more evident from the comparison of the temperatures related to the onset of the thermal degradation (i.e., temperature corresponding to a weight loss of 5 wt.% (T_5%_)) data shown in Table 5. Obviously, T_5%_ decreased with rising TBC loadings, whereas the maximum loss of plasticizer was evidenced on DTG curves at temperatures in the range of 300–310 °C. Still, according to these results, to limit the loss of plasticizer by volatilization at high temperature (NB: this zone is marked by a rectangle in Figure 7b), the processing of plasticized composites must be preferably realized at temperatures that do not exceed 200 °C.

#### 3.1.4. Mechanical Properties

The main goal of plasticizer addition is to increase the ductility and tensile/impact toughness of PLA–AII composites at the level required by application. Regarding the effectiveness of TBC as plasticizer for PLA, it is important to remember that the solubility parameter of PLA (20.2 MPa^1/2^) is very close to that of TBC (18.7 MPa^1/2^) [70], which explains their good compatibility. The evolution of tensile and impact properties of composites with/without plasticizer is shown in Figure 8a–c, whereas the typical profile of stress/strain curves at different amounts of TBC is depicted in Figure 9. Since a low molecular weight plasticizer behaves like a solvent when mixed with a polymer, it leads to decrease of the cohesion between the macromolecular chains, and, consequently, to the reduction of tensile strength [71,72]. The increase of TBC from 5% to 20% into composites led to the gradual decrease of tensile strength at break (Figure 8a), e.g., from 44 MPa for the composite without plasticizer (PLA–AII), to 22 MPa and 14 MPa, respectively, with 10% and 20% TBC in composites. Somewhat surprisingly, the addition of 10% TBC led to the spectacular increase of the strain at the break (ε_b_) to above 160%, whereas when more plasticizer was added, ε_b_ was only moderately changed to 180–190%. On the other hand, the samples without plasticizer (PLA–AII) are fractured during tensile testing without significant plastic deformation/elongation, having an ε_b_ of only 3%. Aliotta L. et al. reported on the combination of rigid fillers (CaCO_3_) with plasticizers (ATBC) in enhancing both the stiffness and toughness of PLA [73]. The plasticizer favoured the plastic deformation of the PLA matrix. The area under the stress-strain curves was defined as “tensile toughness”, a measure of the ability of materials to absorb energy where they are pulled apart or stretched during tensile testing. Accordingly, by considering the area under the stress–strain curves displayed in Figure 9, a marked increase in tensile toughness was seen for PLA–AII–10TBC composites (at 10% plasticizer) compared to the brittle composite (PLA–AII) used as reference. Nevertheless, a good combination between tensile strength and ductility is the key of the toughness [16]. By contrast, at increased amounts of TBC (i.e., 15–20%), only a slight modification in ductility was noted, and not in tensile toughness, because the tensile strength of highly plasticized composites is much lower. At high TBC percentage the saturation with plasticizer is still open for discussion, an assumption supported by ulterior observations that proved the migration of TBC to the surface of specimens after a few months.

Concerning the evolution of mechanical rigidity, the unplasticized composite had the highest Young’s modulus, whereas a moderate, but significant decrease (of about 20%) was obtained by addition of 10% TBC (PLA–AII–10TBC sample). Moreover, by increasing the loading of plasticizer at 15–20%, a typical elastomeric behavior was seen on the profile of stress/strain curves, maybe due to saturation with plasticizer, while the rigidity was dramatically diminished. Still, Figure 10a shows the specimens recovered after tensile testing, proving the differences between the starting (brittle) composites, characterized by low ductility, and the plasticized samples, having much higher elongation.

Regarding the impact properties, PLA is a brittle polymer with low crack propagation energy (measured by notched impact test) and, therefore, it fails by crazing [16]. In several cases, it was reported that the addition of rigid fillers could have positive effects on some polymeric matrices. This enhancement was also stated for PLA composites filled with 20% AII, usually characterized by higher impact resistance than the neat PLAs [26]. On the other hand, from Figure 8c it was evident that low amounts of plasticizer could significantly enhance the impact strength of PLA–AII composites, and, therefore, it might be interesting to use these composites in applications where medium to high impact performances are required. Practically, the addition of 10% TBC significantly improved the Izod impact resistance of PLA–AII composites (i.e., 1.5–2 times), while at higher loadings (15–20% TBC), the specimens showed a rubber-like behavior, and they were not broken (NB). Representative pictures of specimens after impact testing are shown in Figure 10b. 

It is also worth mentioning that additional SEM analyses were performed on the surfaces of fractured specimens after tensile and impact testing, to obtain more information about the morphology and effects of components from composites (short comments and SEM images shown in the Appendix A, respectively). By plasticizing, a more ductile fracture was revealed, especially in tensile tests, with specific features connected to the presence of elongated fibrils and stretched zones formed by the plastic deformation of the PLA matrix during mechanical solicitation (also the SEM images shown in the Section 3.3.1 should be considered).

Interestingly, the highly plasticized composites could be considered “super-tough” PLA materials because they were not broken by impact solicitation (hammer speed of 3.46 m/s). Unfortunately, due to their lower stiffness, they could easily change their form by flexion/mechanical deformation at RT, behavior which is not desired for many products.

To summarize, the overall characterizations allowed us to conclude that “green” TBC is an effective plasticizer for PLA/AII composites. The results agreed with those previously obtained by other scientists for different PLA systems that highlighted the efficiency of citrate plasticizers. By comparing with the unplasticized composites, the sample PLA–AII–10TBC, containing only 10% plasticizer, showed unexpected properties: adequate thermal stability, good tensile strength and moderate rigidity, high ductility (ε_b_ of 160%), good impact resistance (5.6 kJ/m^2^). Still, it is important to point out that the phenomenon of migration of the plasticizer during aging was less evident for this composition compared to the samples with increased plasticizer amounts (15–20%). Therefore, it was decided to optimize and produce this composition in larger quantities as new (low cost) plasticized PLA composites characterized by improved toughness and ductility.

### 3.2. Tailoring the Melt Viscosity of Composites by Reactive Blending

The goal of this section was to reveal a promising route to increase the melt viscosity of plasticized PLA–AII composites, and, consequently, to improve their processability for the extrusion process (NB: compromised by the presence of plasticizer). It is important to note that PLA can be processed using traditional techniques, such IM, sheet extrusion, blow molding, thermoforming, and so on. Typical MFR for the extrusion of PLA blends may be 2–6 g/10 min, while the IM requires higher fluidity, i.e., 10–35 g/10 min [74]. PLA needs to be tailor-made to have higher melt viscosity/melt strength, therefore the chain extension and/or branching is typically applied to produce PLA materials more suitable for processing by extrusion, blow molding, and foaming.

Unfortunately, PLA exhibits some limitations when it is compared to engineering or traditional polymers, because of its easy degradation (i.e., by hydrolysis, shear, thermal oxidation, depolymerization due to presence of impurities or additives, and so on), with negative effects on the molecular properties. To overcome some of these drawbacks, and to enhance/increase the molecular weights, mechanical, thermal, and rheological properties, the addition of CEs, such as Joncryl, into PLA blends was considered by different research groups [59,60,61,65]. These CEs are styrene–acrylic multifunctional epoxide oligomers (structure shown in Figure 1) which can react with PLA end groups (–COOH and –OH) [64]. They are tailored with high or mid epoxy functionality, to increase the molecular weights through PLA chains branching for high melt strength, or to lead to moderate extension/branching, for increased processing speed. However, by considering the reaction rates of epoxide with different terminal groups [62], it is assumed that the kinetics of reaction between the epoxy and carboxylic acid groups would be much higher than those with the primary hydroxyl groups. More information and details about CE features and the mechanisms of reaction with different polymers (including PLA [64]) can be found in the scientific literature and patents, but it is far away from the scope of this paper to develop some typical aspects already treated in literature.

Since the efficiency of epoxide–oligomeric CEs have already been reported for PLA and its blends [59,61,75], one goal of the study was to ascertain to what extent the addition of CE into a composition of interest (i.e., PLA–AII–10TBC) enhanced its rheological properties. As was shown in the previous section (Table 2), the melt fluidity of this sample was extremely high (i.e., MFR 19.6 g/10 min), which meant that the plasticizer damaged the properties of processing (e.g., by extrusion), because typically a much lower MFR is required. First, using as equipment the internal mixer for reactive blending, a rapid tuning was realized to evidence the effects of CE addition into PLA, in the presence of TBC as plasticizer or not. For more insight, Figure 11 shows the modification of torque during mixing of a PLA/1 J composition compared to that of PLA plasticized with 10% TBC, and reactively modified with 0.5% or 1% J. The evident increasing of torque during the time of mixing was reasonably ascribed to the increasing of melt viscosity due to the reactions between the CE and PLA end-groups (–COOH and –OH), leading to the extension and/or branching of PLA chains. However, the PLA/1 J sample showed very high viscosity (MFR not measurable at 190 °C) and this was a supplemental indication of CE reactivity with the PLA terminal groups. Joncryl extended/branched the chains of PLA, and this led to the increase of PLA molecular weights, an assumption that could be additionally proved by molecular and rheological analyses [59]. On the other hand, the addition of plasticizer (10% TBC) with lubricating effects in PLA/J blends considerably decreased the values of torque/melt viscosity. Regarding the melt fluidity (MFR), comparing with the unplasticized composition (PLA/1 J), which was very viscous, adding 10% TBC into PLA/1 J and PLA/0.5 J compositions, caused respectively the MFR to slightly increase to 0.7 g/10 min (high melt viscosity) and, more markedly, to 4.7 g/10 min.

Figure 12 shows the comparative evolution of torque of the plasticized composites (30% filler, 10% plasticizer) following reactive blending with up to 1% J. It was observed that the torque, and implicitly the melt viscosity, significantly increased by raising the loadings of Joncryl (0.3%, 0.5% and 1% J), when compared to the reference without CE (PLA–AII–10TBC). On the other hand, the effects of CE (J) at different amounts were clearly evidenced by the decrease of MFR values (Figure 13). Indeed, adding 0.5% and 1% J in blends, induced a spectacular reduction of MFR to, respectively, 3.9 and 1.5 g/10 min. This certified the possibility to tailor/modulate the rheology of plasticized composites (here, PLA–AII–10TBC, initial MFR of 19.6 g/10 min), by considering the type of processing (e.g., extrusion, which requires high viscosity, or IM, which needs medium/low viscosity). At this time, we would not totally deny the hypothesis that the presence of plasticizers or/and of fillers can lead to some changes/delays in the kinetics of reaction between CE and PLA end-groups. Moreover, a possible reaction between the hydroxyl groups of TBC and the epoxy functions [76] of CE, is not totally excluded, but, for instance, we do not have this evidence.

In relation to the results obtained with different CEs from the Joncryl^®^ category, it is important to consider that their reactivity can be very different, whereas the conditions of mixing (residence time, shear, temperature, etc.) are key parameters that carefully require consideration. Moreover, the reactive process is influenced by other important factors, such as the molecular weights and chemistry of PLA, characteristics of CE (e.g., molecular weights and epoxy equivalent), CE percentage in PLA blends, beneficial/negative effects of fillers or additives, among others. It was reported elsewhere that adding multifunctional CE (as in the present study) to PLA blends led to noticeable increase of PLA molecular weights, whereas a non-linear chain extension (chain branching) could be assumed [77].

From the results shown in Figure 13, it was observed that the plasticizer (TBC) dramatically increased the fluidity of PLA melts, while the CE (J) counterbalanced these effects by significantly boosting the melt viscosity, due to the rise of PLA molecular weights [59]. It is also worth mentioning that the approach was confirmed with only a few differences using another PLA grade (more information shown elsewhere). Undeniably, the optimized CE percentage allowed the remarkable enhancement of the melt rheology (melt strength) of plasticized PLA composites for better processing, i.e., by extrusion or thermoforming.

### 3.3. Current Prospects: Plasticized Composites Produced by REX

For further upscaling on pilot plants and confirmation of the results obtained with laboratory measuring mixers, it was decided to use twin-screw extruders (TSE) to produce higher quantities of plasticized composites modified with low amounts of CE. Nevertheless, regarding the differences between the reactive melt mixing with internal mixers and TSE, it is believed that the temperatures of melt compounding, and the shear and residence time, are parameters that can lead to some differences regarding the properties of final products. For a simplified reading and more comprehension, the discussion here focuses on a plasticized composite (PLA/0.3J–AII–10TBC (TSE)), produced in larger quantities and tested with optimistic results to extrude tubes (straws) and films. Practically, PLA was dry mixed with 0.5% J powder (PLA/J weight ratio of 99.5/0.5) and used for REX (details in the Experimental part), whereas the amounts of filler and plasticizer were kept at 30% AII and 10% TBC, respectively. For comparative reasons, hereafter, a composite without plasticizer (PLA–AII (TSE)) is used as reference.

#### 3.3.1. Characterization of Composites Produced by REX

First, it is worth mentioning that the plasticized composites produced without Joncryl (i.e., PLA–AII–10TBC (TSE)) demonstrated high fluidity (MFR about 17 g/10 min), and, therefore, they are mainly recommended for IM applications, and not for extrusion or thermoforming. On the other hand, the addition of Joncryl via REX (0.3 wt.% J in final composition), allowed a significant, but moderate, reduction of MFR (from 17 g/10 min to 12 g/10 min), and this once more proved the effectiveness of REX (even at low percentage of CE) in modifying the rheological properties of PLA composites.

Table 6 shows the comparison of the main properties of PLA/0.3J–AII–10TBC (TSE) and of unplasticized composites (PLA–AII (TSE)). Undeniably, the plasticized composites (NB: specimens obtained by IM) show distinct mechanical characteristics compared to the reference: lower rigidity/Young’s modulus (i.e., a decrease of about 30% of rigidity), high elongation at the break (>110% vs. only 3.5%), and increased impact resistance (4.9 kJ/m^2^ compared to 3.1 kJ/m^2^). The changes of properties are reasonably ascribed to the effective plasticizing of PLA matrix (T_g_ is about 33 °C).

Moreover, the SEM images (Figure 14a–d) performed on samples fractured at different speeds by tensile and impact testing confirmed the higher plasticity/ductility of the plasticized PLA matrix (Figure 14a,c). By contrast, as especially evidenced from the fractured surfaces obtained by tensile testing (Figure 14b), the unplasticized composites showed brittle behavior (the long “fibrils” and elongated plastic regions are missing), whereas there was a better visualization of the interfacial zones between the polymer matrix (PLA) and AII filler. Furthermore, these images evidenced the partial debonding, or even the removing, of AII microparticles from the polyester matrix, following the mechanical stress. However, the hypothesis that by plasticizing the melt viscosity is significantly decreased, with potential effects in better filler dispersion and more intimate contact between components (PLA and AII), is relevant here. Still, the plasticized matrix determined to a great extent the mechanical properties of composites (assumption supported by SEM images, e.g., Figure 14a).

On the other hand, the important decrease in the stiffness by plasticizing was assessed by the flexural tests: the flexural modulus was nearly three times lower than that of unplasticized composites, and also the deformation by bending was much higher, denoting higher flexibility/lower stiffness. Furthermore, in the conditions of flexural testing (maximum bending of 15 mm), only the specimens of unplasticized composites were broken. Besides, as was expected, the addition of plasticizer led to the change of tensile strength (the tensile strength at yield is remaining high enough on specimens produced by IM, i.e., of 38 MPa), whereas the thermal analyses (DSC) confirmed the decrease of T_g_ (i.e., from 62 °C (unplasticized composite), to about 33 °C by plasticizing).

However, regarding the morphology of composites obtained by REX, it is important to note that, in both cases, the distribution of AII through the PLA matrix (in the presence or not of plasticizer) remained adequate, without evidence of the presence of aggregates of microparticles, with only minor differences regarding the better dispersion of AII in plasticized composites (comparative SEM (BSE) images are shown in the Appendix A). Nevertheless, the different conditions of production (use of internal mixer or TSE) and processing (CM or IM) could lead to some changes regarding the final properties of composites, but the overall results (thermal, mechanical, rheological–MFR) were in good agreement. As mentioned before, the sample PLA/0.3J–AII–10TBC (TSE) was produced in larger quantities and tested at laboratory scale to produce IM specimens, tubes (straws) and films (Figure 15), or by potential users specialized in the extrusion of drinking straws.

#### 3.3.2. Characterization of Aged Films

It is important to point out that, contrary to the highly plasticized composites (15–20% TBC), at lower amounts of plasticizer (i.e.,10% TBC in composites), the leaching/migration of plasticizer to the surface of products (films, tubes, IM specimens, etc.) was not observed, even after prolonged aging (i.e., 2 years). Therefore, it was considered of interest to share this information in the frame of present paper, as a kind of additional confirmation for the approach developed in this study.

Table 7 summarizes the comparative results of mechanical and thermal characterizations obtained on films (thickness ~120 µm), where the initial properties are compared to those after 2 years of aging (NB: under room ambient conditions). Regarding the mechanical properties, excepting the change of the strain at break from 200% to 130%, there were no notable modifications of the main characteristics for the aged films. For instance, the quite surprising preservation of mechanical properties after prolonged aging can be ascribed to the initial presence of the plasticizer in the amorphous phase of the PLA matrix and to its initial high crystallinity (about 20%).

Figure 16 shows the comparative DSC curves obtained on the initial, and aged films up to 2 years. A tiny increase of the DC (i.e., from 19.7% to 26.8%) during the aging was found according to the DSC analyses, and this could explain some inherent changes for the mechanical properties (e.g., of the strain at break). Still, the T_m_ remained at similar values (162–163 °C), whereas the T_g_ event was less evident on the DSC curves of aged films, because of the moderate increase/change of crystallinity (T_g_ was measured to be about 30 °C by DSC (at 10 °C/min), and it was also validated by DMA (Dynamic Mechanical Analysis), from the peak of loss modulus).

In relation to the plasticized composites concerned in this section (i.e., PLA/J–AII–10TBC), it was assumed that the increasing of CE loading up to a maximum of 1% (tests not realized, due to missing of this specific PLA grade), would further raise the melt strength/viscosity of the blends produced by REX, with beneficial effects regarding the achieving of lower MFR, to allow their enhanced processing (i.e., by extrusion or thermoforming). Moreover, the approach applied in this work could be extended to other PLA grades. Indeed, peculiar results concerning the rheological properties (i.e., low MFRs), were obtained using a PLA of higher molecular weights (PLA 2003D–NatureWorks, which had an MFR of 6 g/10 min at 210 °C) and adding only 0.3% J to the plasticized composites. For simplicity, this information and short comments are shared in the Appendix A. Therefore, before concluding, it is important to point out that the results of REX are highly dependent on the characteristics of PLA and of CE/Joncryl (its loading), the type of equipment and the conditions of processing. An experimental fine-tuning is highly recommended to find the optimal REX conditions/compositions. However, in the frame of further prospects it would be important to reconfirm the performances of these novel composites and to obtain additional information regarding their behavior under different conditions/temperatures of utilization, or about the crystallization mechanisms, via alternative techniques of investigation (polarized light microscopy, X-ray diffraction, etc.). Still, a comparative investigation concerning the effectiveness of different citrates for the plasticization of PLA–AII composites is missing, and, therefore, a forthcoming study could be of further interest. To shorten the paper, some information was intentionally omitted here, or shown in the Appendix A. For instance, novel PLA–AII plasticized composites were produced by REX and characterized to evidence their lower stiffness, enhanced ductility, and toughness.

## 4. Conclusions

This study answers at current requests regarding the production of mineral-filled PLA composites designed with tailored properties (i.e., improved toughness and ductility) to allow their larger and long-term utilization. First, the effects of a green plasticizer (5–20 wt.% TBC) in a PLA–AII composition reference (30% filler) were deeply evaluated in terms of morphology, mechanical and thermal properties, focusing a great attention on the enhancement of PLA crystallization (following the synergy of plasticizer and filler) and on the decrease of T_g_. The addition of only 10% plasticizer led to plasticized composites characterized by improved tensile and impact toughness. The sample considered for additional development (i.e., PLA–AII–10TBC) showed interesting properties after processing by CM: tensile strength of 22–26 MPa and a ε_b_ of 160%, good impact resistance (5.6 kJ/m^2^). It should be noted that the mechanical tests of composites having higher TBC amounts (15–20%) evidenced typical elastomeric behavior, whereas the migration of plasticizer was seen even after short time of aging. Correlated with the amounts of plasticizer, a dramatic drop of the melt viscosity was revealed during melt blending and, consequently, an increase of melt fluidity (MFR).

Therefore, for applications requiring improved melt strength and increased viscosity (extrusion, thermoforming), the study proposes as novelty the control of rheology by reactive blending with up to 1% Joncryl, used as a multifunctional chain extender (CE). Undeniably, plasticized composites can be formulated with enhanced melt strength/viscosity (low or medium MFR, function of application) by choosing the optimal CE percentage. Furthermore, higher quantities of plasticized composites modified with CE were produced by REX using TSE and characterized, from the point of view of morphology, mechanical, and thermal properties. The plasticized composites produced by REX (i.e., PLA/J–AII–10TBC) displayed improved rheology (lower MFR) and distinct mechanical characteristics: reduced rigidity/Young’s modulus and stiffness/flexural modulus, high elongation at the break, and increased impact toughness. Furthermore, in the frame of current prospects, the new composites have been tested with promising results for the extrusion of tubes (straws) and films. It is important to point out that, contrary to the highly plasticized composites (with 15–20% TBC), the migration of plasticizer was not noted at 10 wt.% TBC. Moreover, the mechanical and thermal characterizations of films after 2 years of aging did not evidence any dramatic change in properties.

## Data Availability

Not applicable.

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
