# Peer review of "Tailoring and Long-Term Preservation of the Properties of PLA Composites with “Green” Plasticizers"

_polymers, 2022, doi:10.3390/polym14224836_

Round 1

Reviewer 1 Report

Modern industry sets the task of creating materials with strictly specified mechanical properties, stability or, conversely, the ability to degrade, to be economical and safe ones, etc. Composite materials including those based on biodegradable polymers are promising in this regard. The introduction of certain components, for example, clays, minerals, leads, unfortunately, not only to an improvement in a number of properties, but also to losses, for example, in strength. In this regard, research devoted to the creation of new composite materials, as well as the determination of their properties remains an urgent task of the research.

As an original solution to the problem of reducing the ductility and increasing the plasticity and impact resistance of a composite material containing polylactide and CaSO4 β-anhydrite II form, the authors propose to introduce tributyl citrate as a plasticizer. To overcome the dramatic reduction in melt viscosity due to plasticization, the study proposes the addition of a reactive multifunctional chain extender (Joncryl®) as a major innovation. Such a composition of the material is proposed for the first time.

Further, the authors devoted considerable attention to a thorough description of both the methods for production of the composites and their analysis and testing using modern approaches and equipment. The results of the study of the composites with different ratios of the components made it possible to select those that meet the requirements of the industry, including the manufacture of pipes and films obtained by extrusion, and offer them for further development and production on a larger scale. Thus, the tasks set have been generally solved and leave room for further improvement of this class of materials.

The references seem to be appropriate.

The scales in Fig. 5 and Fig. 14 must be more legible.

Reviewer 2 Report

In this research, the article author represented one of the growing areas in polymer processing for different applications based on plasticization that has high industrial translational potential. To improve scientific and technical aspects I have a few comments and with comments, I strongly recommend accepting it.  

1. Author analyzed the cost of industrial translation for different applications as the cost of PLA monomer is very high like 1 kg approximately cost 1000 USD or may be higher. So, do you think it is worth to used except for biomedical applications?

2. What is the major difference you observed between your tributyl citrated based approach and other citrate reported in the literature, superior/inferior/equal?

3. Figure 5. Images should be of good quality.

Reviewer 3 Report

The article is interesting and can be published in its present form

Author Response

Comments and Suggestions for Authors

The article is interesting and can be published in its present form.

Authors: We would like to thank the Reviewer for the time spent in reading and reviewing our manuscript, for the appreciation of results obtained in this study, and finally, for the accord for publication. The authors are hoping that the experimental pathways, the new results, and approaches developed in this work will be appreciated by both categories of scientists, from the academia and industry.

Round 2

Reviewer 2 Report

Dear Author,

Thank you for the wonderful comments. We accept in present form.

Regards,

PSG